# Multi-Element Assessment of Potentially Toxic and Essential Elements in New and Traditional Food Varieties in Sweden

**DOI:** 10.3390/foods12091831

**Published:** 2023-04-28

**Authors:** Barbro Kollander, Ilia Rodushkin, Birgitta Sundström

**Affiliations:** 1Swedish Food Agency, SE-751 26 Uppsala, Sweden; 2Division of Geosciences, Luleå University of Technology, SE-971 87 Luleå, Sweden; ilia.rodushkin@alsglobal.com; 3ALS Scandinavia AB, SE-971 87 Luleå, Sweden

**Keywords:** heavy metals, minerals, cereals, gluten free, algae, seaweed, cadmium, inorganic arsenic, iodine, selenium

## Abstract

With the global movement toward the consumption of a more sustainable diet that includes a higher proportion of plant-based foods, it is important to determine how such a change could alter the intake of cadmium and other elements, both essential and toxic. In this study, we report on the levels of a wide range of elements in foodstuffs that are both traditional and “new” to the Swedish market. The data were obtained using analytical methods providing very low detection limits and include market basket data for different food groups to provide the general levels in foods consumed in Sweden and to facilitate comparisons among traditional and “new” food items. This dataset could be used to estimate changes in nutritional intake as well as exposure associated with a change in diet. The concentrations of known toxic and essential elements are provided for all the food matrices studied. Moreover, the concentrations of less routinely analyzed elements are available in some matrices. Depending on the food variety, the dataset includes the concentrations of inorganic arsenic and up to 74 elements (Ag, Al, As, Au, B, Ba, Be, Bi, Ca, Cd, Co, Cr, Cs, Cu, Fe, Ga, Ge, Hf, Hg, K, Li, Mg, Mn, Mo, Na, Nb, Ni, P, Pb, Rb, S, Sb, Sc, Se, Si, Sn, Sr, Ta, Te, Th, Ti, Tl, U, W, V, Y, Zn, Zr, rare Earth elements (REEs) (Ce, Dy, Er, Eu, Gd, Ho, La, Lu, Nd, Pr, Sm, Tb, Tm, and Yb), platinum group elements (PGEs) (Ir, Os, Pd, Pr, Pt, Re, Rh, Ru, and Pr), and halogens (Br, Cl, and I)). The main focus (and thus the most detailed information on variation within a given food group) is on foods that are currently the largest contributors to dietary cadmium exposure in Sweden, such as pasta, rice, potato products, and different sorts of bread. Additionally, elemental concentrations in selected food varieties regarded as relatively new or “novel” to the Swedish market are provided, including teff flour, chia seeds, algae products, and gluten-free products.

## 1. Introduction

Cadmium (Cd), lead (Pb), mercury (Hg), and arsenic (As) are well known for their toxicity. They are on the World Health Organization (WHO)’s list of the top 10 chemicals of major health concern [1]. These four elements are commonly referred to as “heavy metals,” even though arsenic is a metalloid. Their presence in food, both of natural and anthropogenic origin, has caused concern within the European Food Safety Authority (EFSA) [2,3,4,5]. In addition, there has been increasing interest in the occurrence of silver (Ag) [6], aluminum (Al) [7] (EFSA, 2008), and nickel (Ni) [8,9] caused by the increasing use of Ag as a food preservative agent and Al as a food additive, packing material, and component of drinking water production. Nickel has recently attracted more interest and needs further evaluation according to the EFSA, as it was recently concluded that it may pose a health risk, especially for younger age groups, that is not limited to nickel-sensitive individuals [9]. 

Consumers are protected from high levels of toxic elements in food through regulation by maximum levels (MLs), which are listed in Regulation (EC) No 1881/2006 [10]. However, not all toxic elements or types of foodstuffs are included. The MLs are based on risk to human health, occurrence data, consumption patterns, and global trade. The intention is to keep MLs as low as reasonably achievable (the ALARA principle). This means that the ML for a certain element in a certain foodstuff is set as low as possible, considering the existing levels in that specific food, the global trade, the associated health risk, and the extent to which the food is consumed [10]. MLs, therefore, do not necessarily represent a safe level and are not always safe for all types of consumer groups. An enlightening example of the ALARA principle relates to the MLs for Cd in wild and cultivated mushrooms, which are 0.5 and 0.05 mg/kg wet weight, respectively. The toxicity of Cd is the same, but the levels in wild mushrooms cannot, for obvious reasons, be affected in the same way as in cultivated mushrooms, where soil, fertilizers, and water can be chosen to grow mushrooms with lower Cd levels. Therefore, for wild mushrooms to be allowed on the market, their ML must be higher than in cultivated mushrooms according to the ALARA principle. The strictest MLs apply to foods intended for infants and young children, since children are more vulnerable to toxic substances than adults.

The EU regulation of toxic elements in food [10] is a living document and is continuously updated based on available occurrence data, increased knowledge of toxicity, and new consumption patterns. For example, due to the increasing interest in using different species of algae as foods, there is an urgent need to develop MLs for toxic elements in algae products, and discussions are ongoing in Scandinavia and the EU, as well as globally, in this regard [11,12,13]. Until MLs are included in the general legislation on food and food safety, Regulation (EC) No 178/2002 [14] applies to all types of food, including algae. This regulation stipulates that food must not be placed on the market if it is not safe to consume.

In addition, the availability of standardized analytical methods plays an important role, since no reliable MLs can be established if the levels of corresponding elements cannot be determined accurately and in agreement between the analytical laboratories that perform food control analyses. For example, inorganic arsenic (iAs), one of the most common arsenic types in foods of terrestrial origin, was not regulated until the standard EN 16802:2016 [15] was published in 2016. Subsequently, MLs for iAs in rice and rice cakes were introduced into Regulation (EC) No 1881/2006, and more MLs in other food commodities will follow. 

In an average traditional Swedish diet, the total daily intake (DI) of Cd is 14 µg per person. Cereal products represent 40%, potato products represent 22%, and vegetables represent 14% of Cd exposure in the diet, as shown in Figure 1 [16]. These food groups are also important sources of essential elements like cobalt (Co), copper (Cu), iron (Fe), potassium (K), manganese (Mn), molybdenum (Mo), sodium (Na), phosphorus (P), selenium (Se), and zinc (Zn). Information about the concentrations of nutritional elements in certain foods is often more widely available than knowledge of toxic elements, partly due to a long tradition of analyzing nutrients in foods for public food composition databases (for example, [17]) and also due to the labeling regulations for nutrients in food (Regulation (EU) No 1169/2011 [18]). Concerning some toxic elements, analyses are performed regularly in relation to the MLs in the regulations [10], and if the MLs are not exceeded, there are few incentives to examine levels of toxic elements any further. For toxicologists working with exposure assessments, accurate data on the levels of toxic elements in certain foods is crucial, as well as when levels are below MLs. This also applies to nonregulated elements of concern, both for exposure assessments and for the evaluation of any needs for upcoming regulations. However, analyses are often expensive, and in order to reduce costs, the number of samples is often minimized by preparing composite samples that represent certain food groups, instead of analyzing each food item individually. One example of using composite samples is in market basket surveys (MB), which are performed at the Swedish Food Agency [16,19], where the composite samples reflect relative consumption patterns based on the sales figures of certain foods in Sweden. For example, in MB 2015, the composite sample “Cereal products” comprised 65 wt % of different sorts of bread, 10% wheat flour, 10% pasta, 6% rice, 4% oat flakes, and the remaining 5% breakfast cereals. These products were homogenized and analyzed as one sample. The analytical results thus provide an estimation of the intake of specific elements from certain food groups in Sweden. A similar study was recently performed by the German Federal Institute for Risk Assessment (BfR, 2022) reflecting exposure to Cd, Hg, Pb, and Ni in Germany [20]. 

Due to the drive to increase the share of plant-based foods and foods of marine origins, including algae, in the human diet, the variety of “new” and so-called “novel” foods available on the market has expanded. In the EU, this segment is classified under Regulation (EU) No 2015/2283 [21], which states that food items that were not consumed in the EU in substantial amounts before 1997 are classified as novel foods. For example, the brown algae *Sachharina latissima* (sugar kelp) was consumed in the EU before 1997 and thus is not classified as a novel food. Although knowledge of the elemental composition of new and novel foods is crucial for providing scientifically based recommendations for maintaining a healthy diet, for obvious reasons, information on the elemental composition of novel food is scarcer compared to traditional foods. This is also the case for foods that are not classified as novel but are still “new” to the average consumer. 

In response to the concerns of the EFSA and its call for data, especially regarding Cd [2,22], since 2014, the Swedish Food Agency has facilitated data collection on Cd and other elements by analyzing around 100 individual food items yearly. These are mainly traditional foods that contribute significantly to cadmium intake in the Swedish diet, but foods that are newer to the market are also included. 

The aim of this study was to improve knowledge of levels of Cd and other elements within and between individual food items. We also considered concentrations below the current MLs. Our focus was on the food categories that contribute the most to the exposure to Cd in Sweden and those that are readily/broadly available for regular consumers, including both Swedish-produced and imported foods. The data include information on levels of known toxic (As, Cd, Pb, Hg) and essential (Mo, Fe, Cu, Zn, Se, calcium (Ca), magnesium (Mg), etc.) elements, as well as less routinely analyzed elements in food, such as Ni, thallium (Tl), uranium (U), vanadium (V), rare Earth elements (REEs), and platinum group elements (PGEs). For the sake of brevity, only selected results are evaluated and discussed in detail, while the complete dataset is provided to facilitate research by other groups (for example, for estimations of nutrient intake and/or exposure to toxic elements), as well as general discussions on levels of different elements in both traditional and “new” foods.

## 2. Materials and Methods

### 2.1. Reagents and Standards

All calibration and internal standard solutions were prepared by serial dilutions of single-element stocks (1000 and 10,000 mg L^−1^, SPEX Plasma Standards, Edison, NJ, USA and Promochem AB, Ulricehamn, Sweden). Analytical grade nitric acid (Merck, Darmstadt, Germany) was used after additional purification by sub-boiling distillation in a quartz still. Analytical grade ethylenediamine tetraacetic acid disodium salt dihydrate (EDTA), Triton-X100 surfactant, hydrochloric acid, and 25% SupraPure grade ammonia solution (all from Merck) were used as supplied. The three-stage water purification procedure consists of ion-exchange (SeraDest), a Milli-Q system (>18 MΩ; Millipore Milli-Q, Bedford, MA, USA) and finally sub-boiling distillation in Teflon stills (Savilex Corp., Minnetonka, MN, USA). Water produced in this fashion was used exclusively for the dilution of samples, blanks, and standards. For other, less critical operations, water from the Milli-Q system was used directly. 

Reagents and standards used at the Swedish Food Agency were of analytical quality (p.a.) or better. Calibration standards for total determination were prepared by serial dilutions of multi-element stock (Spectrascan, Teknolab AS, Ski, Norway) or for iAs a single-element stock was used (Inorganic Ventures, Christiansburg, VA, USA). Analytical-grade nitric and hydrochloric acids (Merck, Darmstadt, Germany) were used after additional purification by sub-boiling distillation in a quartz still. Ultrapure water (Q-POD Element, Merck Millipore, Darmstadt, Germany) was used for preparation of standards and dilution of concentrated acids, and for other purposes water from a Milli-Q system (>18 MΩ; Merck Millipore) was used. 

Sintering agent was prepared by mixing 45 g sodium carbonate (Na_2_CO_3_; Merck) and 30 g zinc oxide (ZnO, p.a.; Riedel-de-Haen, Seelze-Hannover, Germany) in clean 125-mL low-density-polyethylene (LDPE; VWR International, Stockholm, Sweden) containers.

### 2.2. Instrumentation

Food samples were homogenized using either a Retsch Ultra Centrifugal Mill ZM 100 (Retsch, Haan, Germany), which includes a stainless-steel pan and a 4 mm Ti sieve, or a large food processor (Foss Homogenizer 2094, A/S Foss, Hillerød, Denmark) with a stainless pan and stainless knife, for dry samples. For the homogenization of fresh samples, a Braun food processor with a Ti blade (Braun GmbH, Kronberg, Germany) was used. 

Samples of wild and farmed seaweed were freeze-dried and homogenized at Lund University, Sweden (Freeze Dry System LYPH LOCK 12 (Labconco, Kansas City, MO, USA) and Retsch Grindomax GM 200 (Retsch, Haan, Germany), or at Chalmers University of Technology, Gothenburg, Sweden (FreeZone 8L Benchtop freezedryer, Labconco, Kansas City, MO, USA, and Meat grinder Model C-E22N, La Minerva, Bologna, Italy).

Total digestion was performed by microwave digestion (MARS-5 or MARS-6, CEM Corporation, Matthews, NC, USA) or by graphite hot-block digestion (DHB R3, ALS Global, Vancouver, BC, Canada). 

For the determination of halogens, samples were sintered in a calibrated Heraeus KM 170 furnace (Karlsruhe, Germany). A microwave digestion unit (MARS-5, CEM Corporation) equipped with 12 perfluoroalkoxy (PFA)-lined vessels (ACV 125) with safety rupture membranes (maximum operating pressure 1380 kPa) was used for acid digestion. 

The determination of total concentrations of elements in the majority of samples was performed by high-resolution inductively coupled plasma mass spectrometry (HR-ICP-MS (ELEMENT XR, Thermo Scientific, Bremen, Germany)) at ALS Scandinavia AB, Luleå, Sweden. Samples of wild *Ulva*, pumpkin seeds, sunflower seeds, peanuts, and vegetarian protein products were analyzed at Swedish Food Agency, Uppsala, Sweden, using a single quadrupole ICP-MS (Agilent 7700x ICP-MS, Santa Clara, CA, USA). Levels of inorganic arsenic were determined at the Swedish Food Agency by using strong anion exchange HPLC-ICP-MS (high-performance liquid chromatography–inductively coupled plasma mass spectrometry, Agilent 1260 Infinity Quaternary LC (Santa Clara, CA, USA) and Agilent 7700x ICP-MS). 

### 2.3. Samples

#### Selection of Samples

According to an earlier market basket study [16], food groups such as cereal products, potato products, and vegetables together contribute around 76% to the exposure to Cd in Sweden. Food items representing these groups were selected with a focus on (1) foods with lesser-known levels of Cd, (2) frequently consumed foods, (3) foods containing several ingredients with different maximum levels, and (4) foods that are “new” to the Swedish market for which consumption is increasing or is predicted to increase. In addition, the simplicity of sample handling (i.e., collection, storage, and sample preparation) was taken into account, as well as the availability of samples from other ongoing projects at the Swedish Food Agency. Table 1 provides information about selected food items together with the sampling year and other projects involving the same samples.

### 2.4. Sample Preparation

#### 2.4.1. General

The food items were purchased from major grocery chains in Sweden. At least 1 kg, or a minimum of three packages of each product, was collected following the regulations for sampling and analysis for the official control of levels of trace elements and processing contaminants in foodstuffs (Regulation (EC) No 333/2007 [28]. In general, the entire sample (edible parts) was milled to a fine powder/slurry before a subsample was extracted for analysis. Mills and mixers generally included titanium (Ti) parts in order to avoid contamination from stainless steel, i.e., by Fe, Ni, and chromium (Cr). Liquid samples were thoroughly mixed in a beaker before the subsample was extracted. Any exceptions to this general procedure are described below.

#### 2.4.2. Bread

In order to reduce the amount of soft bread, crisp bread, or rice cake that required homogenization to provide a representative sample, one of two homogenization methods was used. Either each slice of bread in the package was halved and homogenized (the other half was discarded), or every second slice of bread was homogenized (the other slice was discarded). 

#### 2.4.3. Potatoes

Fresh potatoes were washed and scrubbed with ultrapure water in a similar way to that done prior to household cooking. From the potato packages, each potato was divided in half, and one of the halves was peeled and the other was left unpeeled. The peeled halves and the unpeeled halves were mixed separately to form homogenous slurries, representing two subsamples of each sort of potato purchased. This procedure was used to mimic a regular consumer either peeling the potatoes or not prior to cooking and consumption. 

#### 2.4.4. Algae and Algae Products

Algae oil capsules were purchased from health food stores in Uppsala and Stockholm and also from the internet—one package of each. The oil was analyzed separately as two capsules from each product. Algae salad and macroalgae sheets, purchased from local stores, were homogenized in a food processor.

*Ulva* samples were collected, freeze-dried (homemade device) and milled by mortar and pestle at the Department of Marine Sciences, University of Gothenburg, Sweden. The collection sites were located along the Swedish coast and in the south of Norway, as marked in Figure 2 [25]. Samples of sugar kelp were collected from sea farms in the Koster Archipelago on the west coast of Sweden. All samples of wild and farmed seaweed were freeze-dried and homogenized [26,27].

### 2.5. Analytical Procedures

#### 2.5.1. Total Element Determination

The determination of total concentrations of elements in the majority of samples was performed by using HR-ICP-MS at ALS Scandinavia AB, Luleå, Sweden. Sample preparation was performed in a Class 10,000 clean laboratory areas by personnel wearing clean room attire. The general precautions detailed by Rodushkin et al. [29] were taken to minimize contamination. In order to achieve the lowest possible detection limits and to avoid the contamination risks associated with the additional homogenization of samples, the sample amount was increased to >1 g per digestion. Weighing was performed directly into 50 mL acid-washed plastic vessels. After the addition of concentrated nitric acid (10:1, v/m), samples were left to react overnight, followed by graphite hot-block digestion (105 °C, 2 h). Though the complete digestion of refractory phases in some matrices may require the addition of HF acid [30], as elements associated with such phases are unlikely to contribute to the nutritional uptake, the use of this acid was omitted. After cooling, the volume of transparent digests was adjusted to 40 mL with MQ water. Prior to the analysis stage, samples were further diluted to provide a total dilution factor of approximately 100 and a nitric acid concentration of 1.4 M. A set of preparation blanks, duplicate samples, and control materials was prepared alongside the samples. For the determination of halogens, samples were prepared using Na_2_CO_3_ + ZnO sintering [31].

The concentrations of elements of interest were measured by HR-ICP-MS using a combination of internal standards (indium (In) and lutetium (Lu) when REE was not analyzed) added to all solutions at 1 µg/L and external calibration with set of standards matching the acid strength of sample digests. The all-PFA introduction system, high-sensitivity X-type skimmer cone, and FAST autosampler (excluding the contact of sample digests with peristaltic pump tubing) allow an instrumental sensitivity in excess of 2000 counts/s for 1 ng/L In and background equivalent concentrations for ultra-trace elements (Cd, Pb, As) below 0.2 ng/L. Methane was added to the plasma to decrease the formation of oxide-based spectral interferences, improve the sensitivity of elements with high first-ionization potentials, and minimize matrix effects [32]. Spectral interferences were either avoided by using high-resolution MS settings or mathematical correction was performed (for example, isobaric interferences from tin (Sn) and In, as well as molybdenum oxide interferences on Cd isotopes). Concentrations of halogens were measured in a separate sequence using ethylenediaminetetraacetic acid disodium salt dihydrate, Triton X-100, and an ammonia mixture in order to reduce carry-over in the introduction system and suppress the effect of the oxidation state on the analyte intensity [31]. The method detection limits (defined as 3 times the standard deviation of analyte concentrations measured in a set of preparation blanks) are presented in Table 2. The combined, extended (k = 2) measurement uncertainty was below 15% for Co, Cr, Cu, Fe, I, K, Mn, Mo, Na, P, Se, and Zn and between 10 and 50% for As, Ag, Al, Cd, Hg, Ni, and Pb, depending on the element and its level of concentration. This method is based on the accredited method used by ALS Scandinavia AB in their routine work for the analysis of biological matrices [33,34]. 

It should be noted that, because of the numerous unresolved spectral interferences affecting isotopes of some ultra-trace elements (germanium (Ge), osmium (Os), palladium (Pd), rhodium (Rh), ruthenium (Ru)), the accuracy of the analytical results (though subjected to mathematical corrections) for these analytes may have been affected and should be treated with caution [35].

Elements in food items analyzed at the Swedish Food Agency (leafy vegetables (2014), seeds, grains, peanuts (2018), and vegetarian protein products (2019)) were determined by using ICP-MS in accordance with the procedures of the NMKL (no. 186 and EN 15763). Collision gas (helium) was used in order to reduce molecular spectral interference. The method includes total microwave acid digestion of the samples using a mixture of concentrated nitric acid and hydrochloric acid (6 + 1 mL). The method is accredited for foodstuffs by SWEDAC in accordance with ISO/IEC 17025. The detection limits are listed in the Appendix A for samples analyzed with this method (wild *Ulva*, Appendix A; pumpkin seeds, sunflower seeds, and peanuts, Appendix A; vegetarian protein products, Appendix A), along with the expanded measurement uncertainty for each element.

#### 2.5.2. Inorganic Arsenic (iAs) Determination

The determination of iAs in selected food items (algae, bread, gluten-free bread, rice, and rice products) was performed according to the European Standard EN16802 at the Swedish Food Agency. The method is based on extraction with dilute nitric acid and hydrogen peroxide in a hot water bath followed by analysis with strong anion exchange HPLC-ICP-MS. This analytical method was accredited in accordance with ISO/IEC 17025 by the Swedish board for the accreditation and conformity assessment (SWEDAC) of iAs for rice, rice products, and other foodstuffs within the range 1–25,000 µg/kg. The LOD was between 1 and 3 µg/kg, depending on the dilution of the sample prior to analysis. The expanded measurement uncertainty was +/− 19% (coverage factor k = 2) and was estimated based on the reproducibility in collaborative trials within the European Committee for Standardization (CEN), the laboratory’s own results in proficiency tests, and from the analysis of certified reference materials (for more details, see Kollander et al., 2019 [36]). 

### 2.6. Internal Quality Control

Both laboratories routinely use methods accredited by the Swedish accreditation body, SWEDAC, Borås, Sweden, which require comprehensive documentation and control of quality of all the steps from sample preparation and analyses to reporting results. The accreditation demands yearly participation in proficiency tests in order to control and maintain the quality performance of the laboratories. In addition, the Swedish Food Agency laboratory is appointed by the European Commission (EU) to be the National Reference Laboratory of Sweden for elements and nitrogenous compounds for food and feed and yearly participates in proficiency tests (PTs) arranged by the European Reference Laboratory for metals and nitrogenous compounds (EURL-MN) [5] for total elements and inorganic arsenic.

For quality control at ALS Scandinavia AB, a set of matrix-matched certified reference materials (CRM) were used: NIST 1567a Wheat Flour, NIST 1549 Non-Fat Milk Powder and NIST SRM 1547 Peach Leaves (all from the National Institute of Standards and Technology, Gaithersburg, MD, USA); ERM BB184 Bovine Muscle, ERM BB186 Pig Kidney and ERM BB 422 Fish Muscle (all from the Institute for Reference Materials and Measurements, Geel, Belgium); TORT-1 Lobster Hepatopancreas and DORM-4 Fish Protein (both from the National Research Council of Canada, Ottawa, ON, Canada); GBW 07605 Tea (the National Research Centre for Certified Reference Materials, Beijing, China). Selection of CRMs for a given preparation/analysis run depends on the matrix of food samples and expected analyte range. At least one CRM and two preparation blanks were prepared and analyzed for each batch of 36 samples. A criterion of acceptance was analyte recoveries within the 90–110% range for accredited elements and 80–120% for elements when only information or indicative values were available.

At the Swedish Food Agency, certified and in-house reference materials are routinely analyzed and evaluated together with samples for the continuous control of the quality of the analyses, both on a daily and a long-term basis. Examples of CRMs used in this study are NIST 1567a Wheat Flour, NIST 8436 Durum Wheat Flour, NIST 1568a Rice Flour, NIST 1568b Rice Flour, NIST 8415 Whole Egg Powder and NIST 8414 Bovine Muscle Powder (National Institute of Standards and Technology). 

### 2.7. Evaluation of Data

The data were evaluated and compiled at the Swedish Food Agency, and a summary of the Cd results is presented in Table 3. For more detailed results of the distribution of all measured elements in the food matrices studied, see Appendix A. A principal component analysis (PCA, Unscrambler 7.5) was performed on standardized data for pasta and rice by Dr. Jean Pettersson from the Department of Chemistry, BMC, Uppsala University, Uppsala, Sweden.

## 3. Results and Discussion

The concentrations of elements in traditional and “new” foods are summarized in Appendix A. It should be noted that the concentrations of a few elements in some of the samples have been published previously [16,17,23,24,25,26,27,37], but the dataset provided in this study covers many more food matrices and has far greater element coverage.

For example, in Appendix A, the levels of 56 additional elements (Au, B, Ba, Be, Bi, Br, Ca, Ce, Cl, Cs, Dy, Er, Eu, Fe, Ga, Gd, Ge, Hf, Ho, I, Ir, K, La, Li, Lu, Mg, Mn, Mo, Na, Ni, Nb, Nd, Os, P, Pb, Pd, Pr, Pt, Rb, Re, Rh, Ru, S, Sb, Sc, Se, Si, Sm, Sn, Sr, Ta, Tb, Te, Th, Ti, Tl, Tm, U, W, V, Y, Yb, Zn, Zr) are presented together with previously published essential (Cr, Co, Cu, Fe, Mn, Mo, Na, Se, Zn) and toxic (Ag, Al, iAs/total Arsenic (tAs), Cd, Hg, Ni, Pb) elements in MB samples [16]. Levels of elements in the food categories contributing the most to the intake of Cd (potato products, cereals, and vegetables) are also repeated in Appendix A to facilitate comparisons with levels in individual food items.

As this study was originally initiated to gain a better overview of Cd concentrations in foodstuffs, the discussion is focused on this element. Levels of some additional elements of EFSA concern (iAs, Ni, and Pb), as well as concentrations of new emerging contaminants (Tl and REE), are touched upon but in a more condensed form. For brevity, only a few examples of possible ways to use this comprehensive dataset are provided. 

### 3.1. Cadmium

Levels of Cd in food items contributing 76% of its dietary intake in Sweden, along with food items that are more or less new to the Swedish market, are presented in Table 3. The traditional food with the highest Cd concentration, 0.393 mg/kg, was spinach and the food with the lowest concentration, 0.002 mg/kg, was rice. 

Algae products, except for algae oil, were found to have the highest mean concentrations of Cd as well as the highest measured values—over 1 mg/kg. Sunflower seeds were also found to contain substantial levels of Cd, having mean and maximum levels of 0.262 and 0.559 mg/kg, respectively. 

**Table 3 foods-12-01831-t003:** Amount of cadmium (in mg/kg food) present in the three food groups contributing to 76% of the intake of cadmium in Sweden and in some “new” food varieties. For individual concentrations, see Appendix A.

Food Group and Food Item	Number of Samples		Cadmium, mg/kg		
		Mean	Median	Min	Max
**Cereal products, MB 2015 ***	** *5* **	**0.026**	**0.025**	**0.022**	**0.033**
Pasta	104	0.034	0.033	0.011	0.072
Rice	63	0.020	0.017	0.002	0.123
Rice cakes	11	0.042	0.045	0.014	0.077
Soft bread	28	0.029	0.025	0.012	0.080
Breakfast cereals	31	0.025	0.027	0.0003	0.060
Gluten-free bread	16	0.023	0.011	0.001	0.099
**Seeds**					
Sunflower seeds	16	0.262	0.233	0.084	0.559
Pumpkin seeds	11	0.007	0.002	<0.002	0.041
**Potato products, MB 2015 ***	** *5* **	**0.025**	**0.025**	**0.016**	**0.031**
Potatoes	20	0.028	0.025	0.013	0.056
French fries, frozen	15	0.045	0.039	0.010	0.078
Crisps **, salted	8	0.113	0.087	0.059	0.252
*For comparison*					
Sweet potato	2	0.008	0.008	0.007	0.009
Sweet potato fries	3	0.009	0.010	0.005	0.0012
**Vegetables, MB 2015 ***	** *5* **	**0.010**	**0.010**	**0.008**	**0.016**
Rucola 2018	15	0.059	0.068	0.014	0.107
Spinach 2014 [24]	15	0.134	0.103	0.041	0.393
Spinach 2015 [24]	6	0.134	0.141	0.054	0.236
Iceberg lettuce 2014 [24]	13	0.014	0.015	0.002	0.034
**New food *****					
Vegetarian protein products [37]	16	0.014	0.012	0.001	0.039
Quinoa	10	0.053	0.048	0.030	0.124
Teff fluor	2	0.019	-	0.011	0.028
Psyllium seeds	5	0.046	0.043	0.016	0.076
Chia seeds	7	0.010	0.007	0.004	0.020
**Algae products**					
Nori	2	0.839	-	0.232	1.446
Wakame	4	0.322	0.240	0.222	0.585
Algae salad	6	1.280	1.259	1.004	1.604
Algae oil	22	0.008	0.004	0.002	0.024
**Macroalgae**					
*Ulva fenestrata*	2	0.048	-	0.026	0.050
*Ulva lactinulata*	3	0.059	0.070	0.028	0.070
*Ulva intestinalis*	5	0.088	0.072	0.053	0.173
*Saccharina iatissima*	4	0.440	0.452	0.365	0.492

* Data for composite market basket samples [16] for comparison. Results for all elements in all MB food categories, see Appendix A. ** Note that crisps do not contain as much water as fresh potatoes and therefore contain higher levels of Cd. *** New foods refer to foods that are relatively new to the Swedish market and for which the consumption in Sweden is increasing or is predicted to increase.

#### 3.1.1. Pasta Products

There are no EU MLs for pasta products per se, but there are values for the wheat used in pasta. Generally, dry pasta products, such as those analyzed here, contain solely durum wheat and/or wheat, and levels of Cd can therefore be compared directly with the ML for durum wheat (0.18 mg/kg) or with the ML for wheat germ (0.20 mg Cd/kg) [10]. For cereal-based foods intended for infants and small children, the ML is 0.040 mg Cd/kg, which applies directly to the products as sold. Cadmium levels in pasta grouped according to the main ingredients are presented in Figure 3. The level of tAs are also shown in Figure 3 for information only (not discussed here). The Cd levels vary within the groups, and both low and generally higher levels can be seen in all types of pasta analyzed, as well as in pasta made from “other” ingredients (spelt flour, pea, and soy protein). None of the pasta was labeled as food for infants and young children, and, thus, the ML of 0.040 mg/kg does not apply. However, as pasta is often eaten by children, it is important to note that in 33 of the 104 pasta products studied, the Cd concentration exceeded 0.040 mg/kg. 

The Swedish pasta analyzed contained about 45–50% Swedish wheat, while the remainder of the wheat was durum wheat produced elsewhere, for example, in Spain or Kazakhstan, and thus the elemental composition also reflects that of the durum wheat produced in other countries. Despite the variable origin of the wheat, it is still possible to distinguish pasta products made from Swedish and Italian wheat by using the PCA performed using concentrations of 18 elements in different pasta products (tAs, Ba, Bi, Cd, Cr, Cs, Fe, Hg, Li, Mo, Pb, Sb, Se, Sn, Th, Tl, U, and W levels, presented in Appendix A) (Figure 4). The one Swedish-produced pasta grouped with the Italian pastas was produced in Sweden but made from Italian durum wheat. The ability of PCA to distinguish between foods produced at different sites (i.e., determine provenance) using their elemental compositions has previously been reported for durum wheat [38], rice [39], wine [40,41], olive oils [42], and seaweed [43]. The higher Cd levels in Swedish pasta in 2014 were discussed with the producer (2015), and that prompted the use of wheat with lower Cd levels in their products, even though all products had Cd concentrations below the ML of 0.20 mg/kg.

#### 3.1.2. Rice and Rice Products

The acceptable Cd concentration is regulated by EU 1881/2006. The maximum levels are 0.15 mg/kg for rice and 0.040 mg/kg in cereal-based foods intended for infants and young children [10]. The Cd level in rice (Figure 5) is generally lower than in wheat (which is reflected by the lower ML). Since the levels of iAs in rice and rice products are of special concern, data for iAs [23] are included in Figure 5. Only one of the rice products was labeled as a food for infants and young children (rice cakes intended for consumption from eight months of age), and for this product, the corresponding MLs for Cd and iAs were not exceeded. The other rice cakes studied had concentrations exceeding the ML for iAs in food intended for infants and young children. In addition, 6 of the 11 rice cakes also had Cd concentrations that exceeded the ML for Cd in cereal-based foods intended for children. Although these rice cakes are not labeled as being intended for infants and young children, rice cakes are a popular snack. Therefore, since 2015, the Swedish Food Agency has advised that rice cakes should not be given to children under the age of six [23,45].

The elemental compositions of the different rice products were evaluated via PCA, and the results are presented in Figure 6. In total, the concentrations of 45 elements (Ag, Al, tAs, B, Be, Bi, Ca, Cd, Co, Cr, Cs, Cu, Fe, Hf, Hg, I, K, Li, Mg, Mn, Mo, Nb, Ni, P, Pb, Rb, Re, S, Sb, Sc, Se, Si, Sn, Sr, Ta, Te, Th, Ti, Tl, U, W, V, Y, Zn, Zr) and iAs were included used. Most of the variation can be described by iAs, together with 22 of the elements analyzed (tAs, Ag, Bi, Cd, Co, Cs, Hg, I, K, Li, Mg, Mn, Mo, Ni, P, Rb, Re, S, Sb, Se, Si, Zr). White rice varieties were grouped according to the type of rice and country of origin, while wholegrain rice comprised one group containing a higher amount of minerals, showing that the PCA of elemental composition can be a powerful tool for distinguishing between types of rice and production sites.

The levels of the 45 elements analyzed, including iAs, in samples of rice products (rice, noodles, breakfast cereals, rice porridge, rice cakes, and rice drinks, *N* = 101), including iAs and As, previously reported by the Swedish Food Agency, 2015 [23], can be found in Appendix A.

#### 3.1.3. Bread Products, including Gluten-Free Bread

Samples of bread and gluten-free bread samples were collected within the work of the Swedish Food Composition Database [17] and analyzed for concentrations of vitamins and minerals as composite samples. In this study, the 50 individual bread were separately analyzed for 64 elements (Ag, Al, tAs, Au, B, Ba, Be, Bi, Ca, Cd, Ce, Co, Cr, Cs, Cu, Dy, Er, Eu, Fe, Ga, Gd, Ge, Hf, Hg, Ho, I, K, La, Li, Lu, Mg, Mn, Mo, Na, Nb, Nd, Ni, P, Pb, Pr, Rb, Re, S, Sb, Sc, Se, Si, Sm, Sn, Sr, Ta, Tb, Te, Th, Ti, Tl, Tm, U, W, V, Y, Yb, Zn, Zr) presented in Appendix A for bread and gluten-free bread, respectively, including results for iAs in seven products.

The levels of Cd are generally higher in wholegrain wheat breads (fiber-rich white bread) than in breads made from only wheat germ (white bread) (Figure 7 and Appendix A). Based on these findings, the Swedish Food Agency conducted a risk assessment study to evaluate whether wholegrain products in general are beneficial for health, despite their higher levels of Cd. Despite the occurrence of elevated Cd concentrations posing a potential risk to the kidneys, the conclusion was that the health benefits associated with the consumption of wholegrain products outweigh the risk [47]. However, other negative health effects associated with the intake of Cd have been reported in the literature. For example, many studies have demonstrated associations between Cd exposure and a decrease in bone density (osteoporosis) (see reviews by Åkesson et al. [48] and Cheng et al. [49]). In general, bread products made from oats or rye, including wholegrain varieties, contain lower levels of Cd. These differences in Cd levels in cereals are reflected in Regulation (EC) No 1881/2006 [10], where the MLs for Cd in oats and rye (0.10 and 0.05 mg/kg, respectively) are below the MLs for durum wheat and wheat.

Variations in the tAs, Cd, Ni, and Pb concentrations in different sorts of bread are presented in Figure 7. Gluten-free breads, especially dark gluten-free breads, have higher levels of Pb and Ni compared to gluten-containing breads, probably due to the increased amounts of ingredients such as nuts and seeds [50]. The elevated As levels are directly related to the addition of rice flour [51]. As rice (median 0.017 mg Cd/kg) is a common ingredient in gluten-free products, the relatively low level of cadmium compared to that found in gluten-containing bread products is not surprising. However, there is a need for a more thorough analysis focusing on the presence of bread within the gluten-free cereal product category, especially products with high levels of Cd. 

#### 3.1.4. Potatoes and Potato Products

The levels of Al, Cd, and Co were significantly higher in samples of unpeeled compared to peeled potatoes (paired *t*-test, *n* = 9, 95% confidence). The relative change in the elemental composition after peeling is presented as the mean for all nine types analyzed (Figure 8). Here, 100% equals the relative mean element concentration for the potatoes before peeling. A mean of <100% indicates that most of the element was present in the peel, while a mean of >100% reveals that the element was present within the potato and not in the peel. On average, peeling the potatoes reduced the levels by −67, −8, and −12%, for Al, Cd, and Co, respectively (Figure 8). No significant differences between peeled and unpeeled potato samples were seen for Cr, Cu, Fe, Mn, Ni, Pb, and Zn. Many samples, both peeled and unpeeled, contained Se, Hg, and As levels below the respective LOD, and, thus, the effect of peeling could not be evaluated.

In Appendix A, the individual results for all 31 elements analyzed (Al, Ag, As, B, Ba, Be, Ca, Cd, Co Cr, Cu, Fe, Hg, K, Li, Mg, Mn, Mo, Na, Ni, P, Pb, S, Sb, Si, Se, Sr, U, V, and Zn) in 36 potato products, including peeled and unpeeled potatoes, French fries, and crisps, are presented. 

Only one sample of sweet potatoes was analyzed, with and without peel. For this single sample, peeling reduced the concentrations of most of the elements, but no general conclusions could be drawn. Note that, even though sweet potatoes (*Ipomoea batatas*) are often used in the same recipes as potatoes (*Solanum tuberosum*), they are not a type of “potato.” 

According to the market basket survey conducted in 2015, potatoes and potato products contribute 22% of the dietary Cd intake [16]. Unfortunately, there is no information about whether unpeeled products were included in the composite samples analyzed. However, the results from this study show that Cd exposure is typically 8% lower when consuming peeled potato products.

#### 3.1.5. Quinoa and Teff Flour

Quinoa is increasingly being used by restaurants in Sweden as an alternative to potatoes, rice, and pasta. Regarding Cd concentrations, the 10 samples of quinoa analyzed yielded mean and median levels of 0.053 and 0.048 mg/kg, which are similar to the levels reported by Bolaños et al. (2016) [52] but higher than the values obtained for potatoes, rice, and pasta in this study. In Regulation (EC) No 1881/2006 [10], quinoa is grouped with rice and wheat bran, having an ML of 0.15 mg/kg, while Teff belongs to the cereals category, in which the occurrence of Cd is lower, so it has an ML of 0.10 mg/kg. Only two teff samples were analyzed in this study, and the concentrations were in the lower range among the analyzed samples: 0.011 and 0.028 mg Cd/kg, respectively. Lower levels of Cd have been reported in teff flour [51,52,53].

The comparison of the levels of tAs, Cd, Ni, and Pb in quinoa with potatoes, wholegrain pasta (Ni not determined), and rice (wholegrain and parboiled) in this study provides a rough estimate of the differences or similarities regarding these known elements of concern (Figure 9). For Cd and Pb, the four food items were found to have similar profiles. Levels of tAs were several times higher in rice than in the other three products. The highest level of Ni was seen in quinoa, followed by rice. Ni was not an element of high concern in Europe in 2014 when the pasta was analyzed, and thus, the concentration of this element was not measured. 

For example, Ni levels in pasta and wheat flours can be found in Cubadda et al. (2020) [54] or, for wholegrain wheat and durum wheat, in Swedish Food Agency, 2015 [50]. The minimum and maximum levels of Ni presented are 0.029 and 0.077, and 0.05 and 0.3 mg/kg, respectively.

In addition to tAs, Cd, Ni, and Pb, the levels of 69 other elements (Ag, Al, Au, B, Ba, Be, Bi, Br, Ca, Ce, Cl, Co, Cr, Cs, Cu, Dy, Er, Eu, Fe, Ga, Gd, Ge, Hf, Hg, Ho, I, Ir, K, La, Li, Lu, Mg, Mn, Mo, Na, Nb, Nd, Os, P, Pd, Pr, Pt, Rb, Re, Rh, Ru, S, Sb, Sc, Se, Si, Sm, Sn, Sr, Ta, Tb, Te, Th, Ti, Tl, Tm, U, W, V, Y, Yb, Zn, Zr) were determined in quinoa (*N* = 10) and teff flour (*N* = 2) (see Appendix A).

#### 3.1.6. Seeds

Samples of peanuts, sunflower seeds, and pumpkin seeds were collected within the work of the Swedish Food Composition Database [17] and analyzed for concentrations of vitamins and minerals as composite samples. In this study, the 32 individual samples were separately analyzed for 17 elements (Ag, Al, As, Cd, Co, Cr, Cu, Fe, Hg, Mn, Mo, Ni, Pb, Se, Sn, V, Zn), see Appendix A. Sunflower seeds and pumpkin seeds are used in Sweden as common ingredients in bread and other food products. Sunflower seeds are well known to contain Cd at high levels [50,55,56], which is reflected in Regulation (EC) No 1881/2006 [10], where they have a higher ML than other seeds (0.50 mg/kg compared to 0.1–0.3 mg/kg, except for poppy seeds at 1.20 mg/kg). The level of Cd in sunflower seed samples varied from 0.084 to 0.559 mg/kg (median 0.233), while levels were much lower in pumpkin seeds (<0.0018–0.041 mg/kg, median 0.002). The pumpkin seed sample with 0.041 mg/kg, was the only unshelled, roasted, and salted sample, while the rest of the samples were shelled, including the sample with the second-highest concentration of 0.010 mg/kg. The Cd levels were within the same ranges previously reported for sunflower and pumpkin seeds purchased on the Swedish market: 0.29 (0.17) and 0.013 (0.011) mg/kg (mean (median)), respectively [33].

Only five samples of peanuts (all salted, one roasted) were analyzed, so no general conclusion can be drawn from the results, which showed Cd concentrations that varied from 0.012 to 0.153 mg Cd/kg (median 0.030 mg/kg). Despite its name, the peanut belongs to the botanical group of seeds, and its ML is 0.20 mg Cd/kg [10].

Both psyllium and chia seeds are often used as alternatives to similar fiber-rich products containing gluten. In this work, five samples of psyllium seeds were analyzed, and mean and median Cd concentrations of 0.046 and 0.043 mg Cd/kg, respectively, were found. The chia seeds had a lower Cd content with a mean of 0.010 and a median of 0.007 mg/kg. Compared with fiber-rich and gluten-containing alternatives, like wheat bran (around 0.1 mg/kg, [50]), both psyllium and chia seeds contain lower levels of Cd. Rubio et al. (2018) [57] and Bolaños et al. (2016) [52] have reported similar levels of Cd in chia seeds (0.01 ± 0.00 mg/kg (*N* = 18) and <0.016 (*N* = 14), respectively) while a broader range, <0.003 to 0.090 mg/kg (*N* = 9), was reported by Gomez et al. (2021) [58].

In addition to Cd, the levels of 72 other elements (Ag, Al, tAs, Au, B, Ba, Be, Bi, Br, Ca, Cd, Ce, Cl, Co, Cr, Cs, Cu, Dy, Er, Eu, Fe, Ga, Gd, Ge, Hf, Hg, Ho, I, Ir, K, La, Li, Lu, Mg, Mn, Mo, Na, Nb, Nd, Os, Ni, P, Pb, Pd, Pr, Pt, Rb, Re, Rh, Ru, S, Sb, Sc, Se, Si, Sm, Sn, Sr, Ta, Tb, Te, Th, Ti, Tl, Tm, U, W, V, Y, Yb, Zn, Zr) were determined in psyllium and chia seeds (see Appendix A).

The comparison of the tAs, Cd, Ni, and Pb concentrations in the seeds analyzed in this study clearly indicates that sunflower seeds have the highest levels of Cd and Ni (Figure 10). Both chia and psyllium seeds were shown to contain higher levels of tAs than the other seeds, while elevated levels of Pb were only found in psyllium.

#### 3.1.7. Rucola

Samples of rucola were collected within the work of Swedish sampling program on contaminants for analysis of nitrate [24]. In this study, the 15 samples were analyzed for Cd and 70 other elements (Ag, Al, As, Au, B, Ba, Be, Bi, Br, Ca, Ce, Cl, Co, Cr, Cs, Cu, Dy, Er, Eu, Fe, Ga, Gd, Ge, Hf, Hg, Ho, I, Ir, K, La, Li, Lu, Mg, Mn, Mo, Na, Ni, Nb, Nd, Os, P, Pb, Pd, Pr, Pt, Rb, Re, Rh, Ru, S, Sb, Sc, Se, Si, Sm, Sn, Sr, Ta, Tb, Te, Th, Ti, Tl, Tm, U, W, V, Y, Yb, Zn, Zr), see Appendix A. For comparison, we included the Cd concentrations in spinach from the above mentioned sampling program. 

The range of Cd concentrations in rucola was 0.014–0.107 mg Cd/kg, with mean and median values of 0.059 and 0.068 mg Cd/kg, respectively. The ML is 0.20 mg/kg, the same as for spinach, while the ML is 0.10 for other leaf vegetables containing less Cd, for example, iceberg lettuce [10]. For comparison, the Cd levels in spinach and lettuce analyzed within the Swedish sampling program for contaminants are presented in Table 3. Among the spinach samples, the Cd concentration exceeded the ML of 0.20 mg/kg in 3 of the 15 samples analyzed in 2014 and in 1 of 6 samples analyzed in 2015 [24], while all samples of iceberg lettuce (*Lactuca sativa*) had Cd concentrations below the ML of 0.10 mg/kg.

#### 3.1.8. Vegetarian Protein Products

There are many alternative protein sources to meat available on the market. If traditional meat meals are replaced with other protein sources of plant or marine origin, the elemental composition of the intake will also change. Appendix A presents the analytical results for eight elements of interest (Cd, Co, Cu, Fe, Mn, Mo, Pb, Zn) found in 19 composite samples of meals based on soy, oats, peas, and mycoprotein [37]. The names of the products (escalope, nuggets, balls, mince, and bacon) indicate an intended use as replacements for traditional meat products. For Cd and Pb, the means were 0.014 and 0.005 mg/kg, respectively, which could be compared to the mean values of 0.003 and 0.002 mg/kg found in traditional meat products that an average citizen in Sweden consumes (Food Category Meat MB samples, [16]). Consequently, consuming these protein products instead of meat products will result in a double or higher intake of Cd and Pb. Although based on a rather limited dataset, these results should be taken into consideration when planning of a future studies on this subject. In a study by Furey et al. (2022) [59], all five samples of natural fungi protein had Cd levels < 0.01 mg/kg, while De Marchi et al. (2021) [60] presented quantification of 4 out of 10 samples of plant-based burgers (0.14–0.15 mg/kg, no LOD given). Although the latter study is indeed important and the conclusion that “the heavy metals were not detected” is true, its information is not usable without the LOD. In addition, in view of the low but important differences in concentrations of heavy metals between meat and vegetarian protein replacement products, the analytical method used in these kind of studies should have quantification levels in the ultra-trace (sub-10^−3^ mg/kg) range or lower.

#### 3.1.9. Algae

##### General

Algae are considered valuable food resources that could be produced more sustainably than other food sources [13]. The increasing interest in algae as a food, and research aimed at optimizing algae production, particularly in the Nordic part of Europe, have drawn attention to the lack of regulations and management tools available to ensure its food safety [11]. Although not yet regarded as traditional food in Sweden, algae are ingredients in certain meals and in food supplements available on the market. 

In this study, levels of iAs and 70 elements (Ag, Al, As, Au, B, Ba, Be, Bi, Br, Ca, Cd, Ce, Co, Cr, Cs, Cu, Dy, Er, Eu, Fe, Ga, Gd, Ge, Hf, Hg, Ho, I, Ir, K, La, Li, Lu, Mg, Mn, Mo, Na, Ni, Nb, Nd, Os, P, Pb, Pr, Pt, Rb, Re, Rh, Ru, S, Sb, Sc, Se, Si, Sm, Sn, Sr, Ta, Tb, Te, Th, Ti, Tl, Tm, U, W, V, Y, Yb, Zn, Zr) were determined in 23 algae products available on the market (Nori, Wakame, Algae salad, Algea oils; see Appendix A) as well as in four samples of sugar kelp (Appendix A) available through collaboration [26,27]. In addition, 10 samples of different wild *Ulva* species from an ongoing project [25] were analyzed and the results of iAs and 20 elements (Ag, Al, As, Cd, Co, Cr, Cu, Fe, Hg, Mn, Mo, Ni, Pb, Se, Sb, Sn, Tl, U, V, Zn) are presented in Appendix A.

##### Cadmium in Algae Products Available for Human Consumption

The samples of nori and algae salad had the highest Cd levels measured, with maximum measured concentrations of above 1 mg/kg. Similar levels (1.42 ± 1 mg/kg, *N* = 8) were seen in Nori in a study by Todorov et al. (2022) [61].

Algae oils were found to have much lower Cd levels than the macroalgae products with mean and median concentrations of 0.008 and 0.004 mg/kg. 

##### Cadmium in Macroalgae—Wild and Farmed

The Cd level in green macroalgae *Ulva* samples varied from 0.026 to 0.173 mg Cd/kg. The lowest values were found in *Ulva fenestrata* and the highest were found in *Ulva intestinalis*. Samples of the latter were collected from the west and east coasts of Sweden, where the salinity varied from 22 at the west coast to 3 g/L at the northern east coast. Similar variations in Cd concentrations within and between *Ulva* species have also been reported by others [61,62].

Four samples of the brown macroalgae sugar kelp (*Saccharina latissima*) were analyzed for 72 elements and iAs in this study. The mean and median Cd concentrations were 0.440 and 0.452 mg Cd/kg, respectively. All four samples were farmed in the Koster Archipelago. 

##### Growing Interest about the Elemental Composition of Algae

The information available regarding certain elements in macroalgae farmed in the seas of Northern Europe and intended for food production has increased substantially recently [11,27,63,64,65,66], as has the interest of the EFSA [12]. Although the focus is not always on the elemental composition per se, awareness of the importance of the elemental composition of algae is growing, considering both toxic elements, such as As, Cd, Ni, Pb, and Hg, and essential ones, such as Cu, Fe, Mo, Se, and iodine (I). Comparing levels of iAs, Cd, Ni, and Pb between the macroalgae analyzed confirms the broad variations in levels, both within and between different species (Figure 11). Cadmium levels in *Ulva* species were generally lower than in the other species analyzed, while the four samples of *Saccharina latissima* showed the most uniform distribution of these elements, probably because they were harvested at the same age and from the same area [26,27]. In the comprehensive dataset published by Duinker et al. (2020) [66], the levels of iAs, Cd, and Pb in sugar kelp were 0.03–0.67, 0.16–3.1, and <0.22–5.7 mg/kg, respectively (Ni was not analyzed), which shows the possible variations within the same species. Broad ranges of concentrations of Cd and Pb, within and between different kinds of edible seaweed, have, for example, also been reported by Todorov et al. (2022, seaweed available on the market in the United States) [61] and Paz et al. (2019, seaweed available on the Spanish market) [67].

Iodine is of special concern since some species of macroalgae are known to contain potentially unhealthy high levels of this element [11]. In this work, the level of I in algae ranged from 15 mg/kg in nori to 2500 mg /kg in sugar kelp. An even broader range of 8–10,000 mg/kg was reported for the same species by Duinker et al. (2020) [66]. 

Due to the large variations between different species of macroalgae, it is of utmost importance to be species-specific when any conclusions are drawn regarding levels of different elements in seaweed. 

The concentrations of several elements, including bromine (Br), niobium (Nb), strontium (Sr), titanium (Ti), V, yttrium (Y), and zirconium (Zr) in the macroalgae nori, wakame, “algae salad,” and sugar kelp can be 100-fold or even 1000-fold higher than in cereals, potatoes, and vegetables (see Appendix A). Consequently, extended consumption of these algae will increase the nutritional intake of these elements with currently “unknown” biological functions. 

### 3.2. Other Elements in Traditional and New Food Products

The growing interest in the elemental composition of foods focuses not only on “new” foods but also on “new” less-studied elements. Many elements are finding new uses in modern technology applications, which has resulted in a dramatic increase in their demand and, consequently, also in their global extraction rate. So-called ‘technology-critical elements’ (TCEs) include most REEs, PGEs, and gallium (Ga), germanium (Ge), In, Nb, tantalum (Ta), tellurium (Te), and Tl. Despite increasing recognition of their prolific release into the environment, their concentration levels in food remain largely unknown.

For example, Tl is an element that might be of concern due to its high toxicity [68,69] with provisional oral reference dose in 10^−5^ mg/kg per day [70], and also due to its availability to plants used for food [71]. The highest levels of Tl in this work were found in quinoa (0.031 (0.028) mg/kg), *Ulva* (0.015 (0.011) mg/kg), sugar kelp (0.0056 (0.0062) mg/kg), and dark gluten-free bread (0.0038 (0.0038) mg/kg). These concentrations can be compared to levels found in the Swedish market basket survey for 2015, where the food categories Meat and Eggs contained the highest levels of 0.0024 and 0.0025 mg /kg, respectively, while the average values for Vegetables and Cereals were 0.0005 and 0.0008 mg/kg, respectively (Appendix A).

The REEs, another group of elements belonging to TCEs, are considered neither essential nor toxic and have unclear biological roles, if any. Potential risks associated with elevated REE exposure are mainly associated with the ability to displace Ca from calcium-binding sites [68]. Although the first papers on REE content in food were already available some 30 years ago (e.g., [72]), reported levels were significantly (by orders of magnitude) higher than in more recent studies [73], which may indicate either a decreasing level of environmental pollution or, more likely, progress in analytical methods for ultra-trace analysis of food matrices. It should be stressed that, in the majority of recent publications, the interest in REE content in various food products mainly stems from either assessment of environmental pollution and/or botanical/geographical origin identification (e.g., [74,75,76,77,78,79,80]). The majority of food products tested in the present study contain undetectable to very low levels of REEs, demonstrating both their scarcity in the environment and the low bioavailability of these elements. The highest REE concentrations were measured for the food category Sweets. This correlated with Ti concentrations and most likely originated from the use of inorganic/mineral food coloring agents (e.g., titanium dioxide containing REEs as an impurity).

## 4. Conclusions

To the best of our knowledge, this study is one of the most comprehensive of its kind, regarding the number of food varieties tested, the element coverage, and availability of data at truly ultra-trace (sub-10^−3^ mg/kg) levels. The concentration data acquired can complement information on inorganic constituents in foodstuffs that are currently available on product labels, in various food nutritional tables, on the homepages of brand name holders, and in other relevant surveys. A detailed comparison of our results with previously published data (where available), though potentially intriguing, was outside the scope of this investigation. 

Based on the concentration data obtained during this study, it is possible to calculate intakes for both essential and toxic elements, assuming typical consumption scenarios, and such intakes can then be compared with either the recommended daily allowance figures or the accepted toxicity thresholds. In the interest of brevity, we intentionally excluded such topics from the present discussion, although the presented results may aid in the understanding of how ongoing and anticipated changes in the diet of the general population could alter the intake of Cd as well as that of other elements, both essential and toxic. It should be stressed that the selection of food varieties tested is somewhat skewed towards the Swedish market and that concentration levels may change in the future as a result of climate change, nutrient deficiencies in agricultural soils, changes in cultivation practices and animal feed composition, and progress in food processing and packaging, to name but a few factors. 

Though interest in inorganic analyses of food products mainly stems from nutritional concerns and the few toxic elements regulated by law, such analyses are increasingly being used in food authentication or provenance studies, as well as in forensics applications, where the availability of reliable information on trace- and ultra-trace-element concentrations is essential.

This study provides levels of elements that are not routinely analyzed in foods and not requested within the EFSA call for data with a focus on the Swedish market. When the general diet is changed, and with the introduction of elements used in modern technology to the food chain, knowledge about the total element composition in various foods is important to provide a long-term perspective on sustainable health practices regarding both traditional and “new” foods. 

## Figures and Tables

**Figure 1 foods-12-01831-f001:**
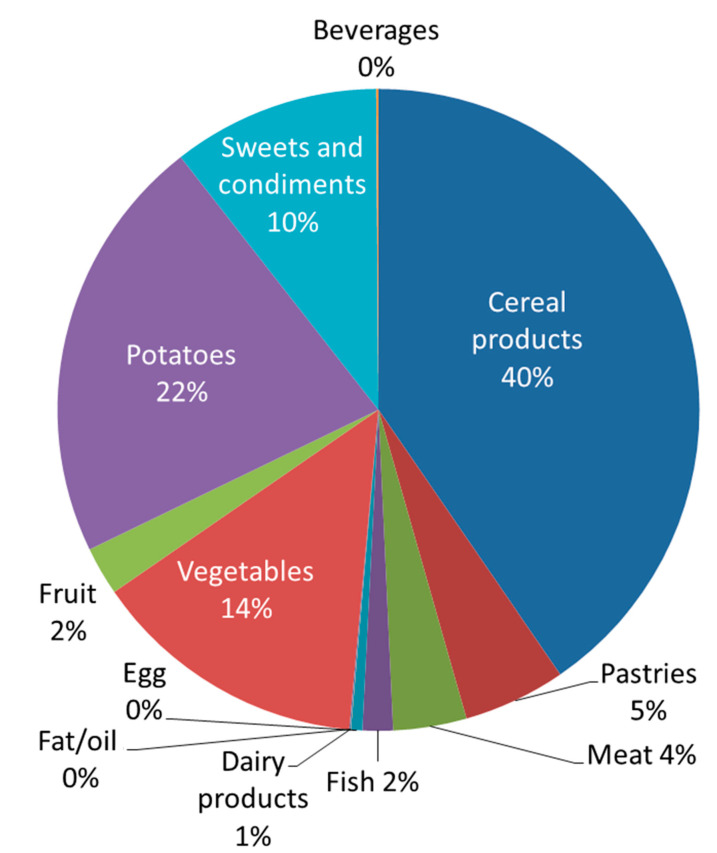
Contributions, as percentages, of different food groups in an average Swedish diet to cadmium intake, based on sales figures. The total daily intake is 14 µg per person. Modified from the Swedish Food Agency, 2017 [16].

**Figure 2 foods-12-01831-f002:**
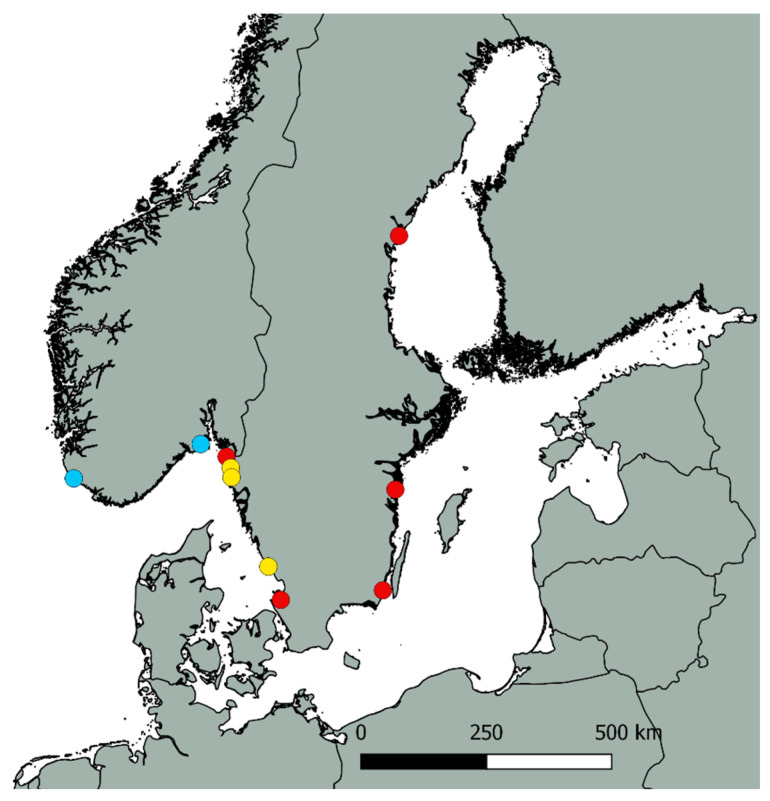
Map showing the sample locations of wild *Ulva* biomass. The blue dots indicate sites at which *Ulva fenestrata* was sampled, the yellow dots show sites at which *Ulva lacinulata* was collected, and the red dots show sites at which *Ulva intestinalis* was collected [25]. All collected populations were examined molecularly for species identification; for GenBank accession numbers, see Appendix A.

**Figure 3 foods-12-01831-f003:**
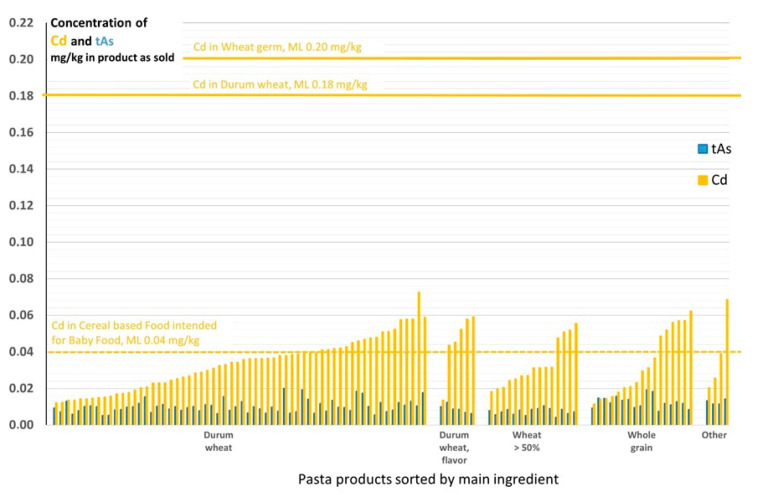
Levels of Cd and total arsenic (tAs) in mg/kg for pasta products, sorted by main ingredient, purchased in Sweden in 2014. The European Union maximum levels (MLs) for Cd are represented by horizontal lines and the dashed lines represent values for foods intended for young children (there are no MLs for As in wheat products). For detailed information about specific products and levels, see Appendix A.

**Figure 4 foods-12-01831-f004:**
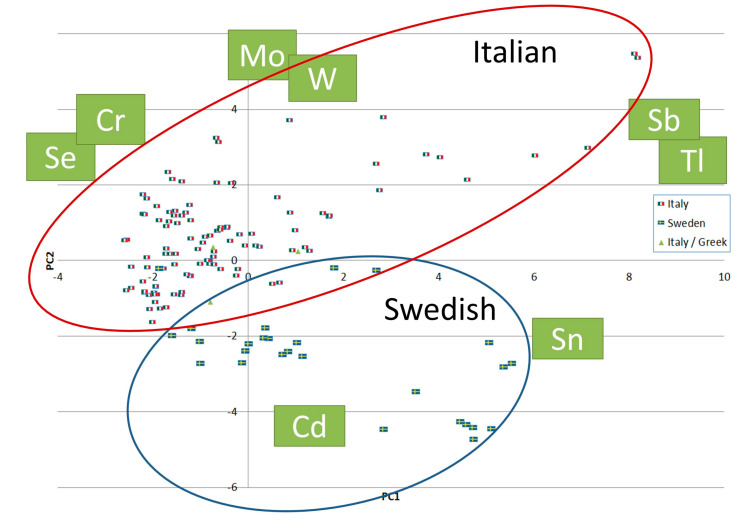
PCA in pasta products purchased in Sweden in 2014. The flags indicate the provenance of the pasta. Pasta products produced in Sweden are marked with two blue circles and those produced in Italy are marked with a red ring. PCA was performed on standardized and centered data from 19 isotopes (Cd111(low-resolution mode (LR)), Cd114(LR), Fe56 (high-resolution mode (HR)), Mo98(LR), Se77(HR), Se78(HR), Sn118(LR), Sn120(LR), Sb121(LR), Sb123(LR), Cs133(LR), Ba137(LR), Ba138(LR), W184(LR), Tl205(LR), Bi209(LR), Th232(LR), U238(LR), and Cr53(HR) [44]. For detailed information, see Appendix A.

**Figure 5 foods-12-01831-f005:**
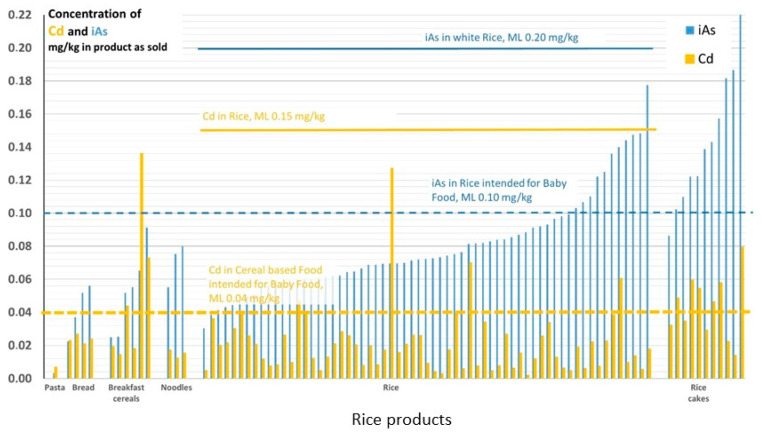
Levels of Cd and iAs in mg/kg for rice and rice products purchased in Sweden in 2015. European Union maximum levels (MLs) are represented by horizontal lines in respective colors and the dashed lines show the MLs for foods intended for infants and young children. Results for iAs taken from reference [23]. For detailed information about specific products and levels, see Appendix A.

**Figure 6 foods-12-01831-f006:**
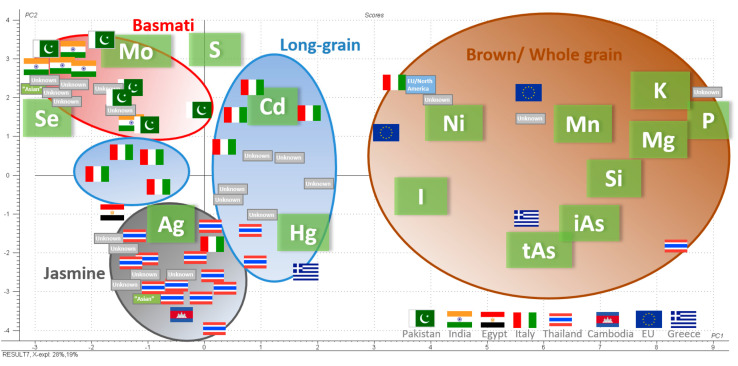
Principal component analysis (PCA) of the elemental levels of 46 elements plus iAs in 63 rice samples. Inorganic arsenic, together with 22 of the elements, describes most of the variation. Each sample of rice is represented by either the flag of its country of origin, the name of the Continent or if the origin is unknown by “Unknown” (according to the package label). Samples of white rice is grouped according to the type of rice and origin, while the samples of wholegrain rice comprises one group containing a higher number of minerals [46]. For detailed information on levels in respective rice sample, see Appendix A.

**Figure 7 foods-12-01831-f007:**
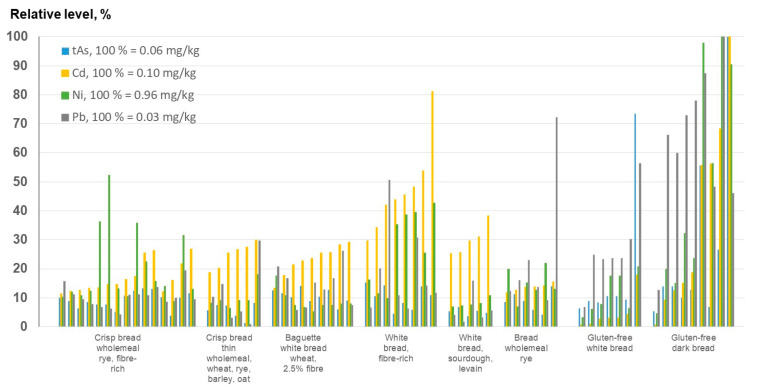
Relative levels of tAs, Cd, Ni, and Pb in different types of bread, including gluten-free bread products. The highest levels of the elements measured in the bread products were set to 100% in order to compare the occurrence of these heavy metals (including arsenic) in each sort of bread. The highest values were found for dark gluten-free products and fiber-rich bread. For detailed information, see Appendix A for bread and S6 for gluten-free bread.

**Figure 8 foods-12-01831-f008:**
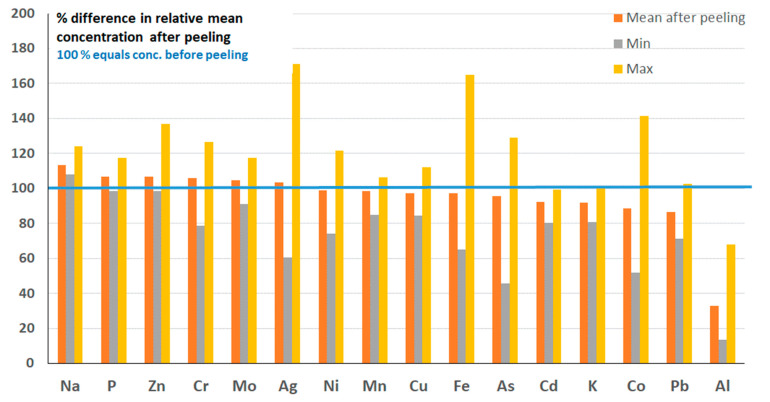
Changes in elemental composition in potatoes after peeling. For information on individual results and additional elements, see Appendix A.

**Figure 9 foods-12-01831-f009:**
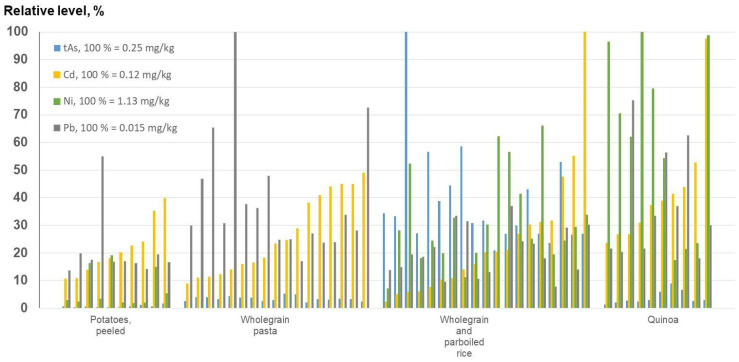
Relative levels of tAs, Cd, Ni (not analyzed in pasta), and Pb in quinoa and products consumed in a similar way in Sweden. For detailed information, see Appendix A.

**Figure 10 foods-12-01831-f010:**
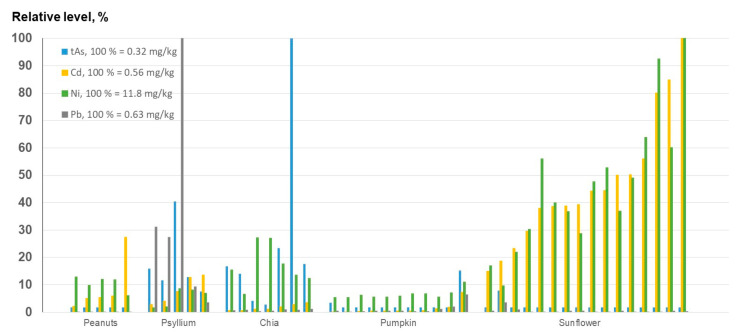
Relative levels of tAs, Cd, Ni, and Pb in seeds. For detailed information and the concentrations of 69 other elements, see Appendix A.

**Figure 11 foods-12-01831-f011:**
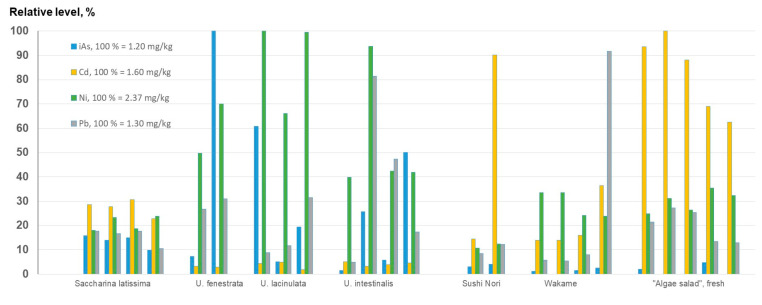
Relative levels of iAs, Cd, Ni, and Pb in macroalgae. For detailed information and information on additional elements, see Appendix A for *Saccharina latissma*, Appendix A for *Ulva*, and Appendix A for algae for human consumption.

**Table 1 foods-12-01831-t001:** Information about the samples analyzed within this work, including the type of food, year of sampling, number of samples, and number of elements analyzed. References are included for projects at the Swedish Food Agency involving the same samples.

Food Group and Food Item	Year of Sampling and Analyses	No. of Samples	No. of Elements Analyzed, Validated/ Indicative	Speciation	If Analyzed for Other Purposes, including Reference
**Cereal products**					
Pasta products	2014	104	8/10		-
Rice products	2015	115	17/ 28	iAs	Inorganic arsenic [23]
Bread	2016	50	16/49	iAs (*N* = 3)	Vitamins and minerals [17]
Gluten-free bread	2018	16	16/48	iAs (*N* = 4)	Vitamins and minerals [17]
**Seeds**					
Pumpkin seeds, sunflower seeds, peanuts	2018	32	17		Vitamins and minerals [17]
**Potato products**					
Potato products	2019	62	16/15		-
**Vegetables**					
Rucola	2019	15	16/55		Nitrate; Swedish sampling program on contaminants [24]
**New food ***					
Vegetarian protein products		17 **	8		Vitamins and minerals [17]
Quinoa, teff flour, chia and psyllium seeds	2021	24	17/55		-
**Algae products**					
Algae oils	2019	11	17/53		-
Algae sold as food	2019, 2021	13	17/53	iAs	-
**Algae**					
*Ulva*	2020	10	20	iAs	Algae for food [25]
Sugar kelp	2021	4	16/56	iAs	Algae for food [26,27]
**For comparison**					
Market basket samples	2015	55 ***	16/56	iAs	Essential and nonessential elements in composite samples [16]

* New foods refer to foods that are relatively new to the Swedish market and for which the consumption in Sweden is increasing or predicted to increase. ** Composite samples of the same food item with different trademarks. *** Composite samples of different food items in the same food group, where five composite samples were analyzed for each of the 11 food groups.

**Table 2 foods-12-01831-t002:** Typical limits of detection (LOD, 3 × std for blank, *n* = 9) with a total dilution factor of 100 for selected elements measured by HR-ICP-MS. Individual LODs for each batch are presented in Appendix A.

Type of Sample	Limit of Detection in the Sample, µg/kg																		
	As	Ag	Al	Cd	Hg	Ni	Pb	Co	Cr	Cu	Fe	I	K	Mn	Mo	Na	P	Se	Zn
Solid	1.4	0.8	46	0.2	1.6	7	1.1	2.6	5	5	27	23	3250	6	0.7	686	498	6	49
Liquid	0.7	0.4	23	0.1	0.8	4	0.6	1.3	2.6	2.6	14	12	1630	2.8	0.4	343	249	3	25

## Data Availability

The data presented in this study are available in Appendix A.

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
