# Peer review of "Multi-Element Assessment of Potentially Toxic and Essential Elements in New and Traditional Food Varieties in Sweden"

_foods, 2023, doi:10.3390/foods12091831_

Round 1

Reviewer 1 Report

I recommend that researchers incorporate:

1. A comparative table between the data of your study and that of other authors.

2. A table visualizing the data that they call the old and the new.

3. Calculate the dietary risk from consumption of the foods studied. 

Author Response

Reply to Reviewer 1.

We warmly thank you for your input. Please, see our replies and comment below.

"I recommend that researchers incorporate:

  1. A comparative table between the data of your study and that of other authors."

We agree that such a table would ineed be interesting but considering the amount of elements and samples analyzed it we believe that would be too large for an article. Also, many of the elements provided here in certain food matrices are have not been seen in the literature before, to our knowledge.

        "2. A table visualizing the data that they call the old and the new."

We have now clarified which elements that have been published before in which samples. 

        "3. Calculate the dietary risk from consumption of the foods studied."

Dietary risks are indeed important and very interesting. However, this was out of the scope of this study which aimed to provide data and for a cadmium, comapre levels between certian foodstuff. The data is provided for others to evaluate dietary risks. 

Reviewer 2 Report

In this study the authors present the data on levels of cadmium and a wide range of elements determined in foodstuffs of plant origin, collected from Swedish market. Indeed the work is really worth and valuable and of course the consumption of these “novel” foods is growing up in the recent years. So these types of studies should be implemented. For this reason I strongly invite the authors to improve this manuscript, to better gather ideas, to develop the sections of the manuscript, and to request the help of a native English speaker for proofreading, because it deserves to be published after several refinements. As it stands, let me say that the manuscript is still in the "draft" stage and needs to be improved.

Key issues:

-        the introduction is really too long and disproportionate to the rest of the work (particularly in comparison of results and discussion)

-        there are too many errors which make it difficult to read the manuscript smoothly

-        the references are indicated in the wrong style and often not indicated at all (I suggest to the authors to re-read with care the journal guidelines)

-        authors should standardize the reference style, check and expand the references used.

-        the style of the conclusion must be changed. It almost feels like a continuation of findings and discussion. You should give first again your objective, then you show your major and most important results and achievements.

-        the presentation of the results and their discussion are insufficient, with such a beautiful dataset you should be able to obtain even more than one article! For example for algae samples the discussion about the content of iAs is only hinted at, while the concentrations of I in these matrices (as well as in the others) it's not really presented. Although I am of the "party" that data often speak for themselves, you cannot think that a dataset alone with an introduction is an article

 Some other problems:

-        for all instrumentations reagents standards (Company, City, Country). Standardize and change it in all manuscript.

- Did you also use “>1 g” of sample for the seaweed? I can't believe it because I know the matrix well and I know that for a complete digestion it is sometimes necessary to add other acids (e.g. hydrofluoric acid...).

-        the “materials and methods” section is not clear at all.

-        check the units of measure of all the text!!! for example: line 332 Are you sure that the limit for cadmium in cereal-based food intended for infants and young children is “0.040 µg/kg”???

-        when using an abbreviation in the text you should introduce it first (e.g. LOD)

-        it would be nice if you compare your results with other recent studies in the literature.

-        iodine quantification: Have you had any memory effect problems? same dilution as the other determinations? or even worse recovery problems?

Once again, I warmly invite the authors to make efforts to improve the manuscript and then resubmit it.

Author Response

Reply to Reviewer 2

We sincerely thank you for the valuable and encouraging comments regarding our manuscript ijerph-2235224 “Cadmium and 73 other elements in foodstuff being major contributors to current dietary exposure of cadmium and in “new food” varieties”. With your help, we have managed to improve it substantially. See our reply below after each of your comments/recommendations.

Best regards,

Dr. B Kollander, Dr. Ilia Rodushkin and Ms Birigtta Sundström

Comments and Suggestions for Authors

“In this study the authors present the data on levels of cadmium and a wide range of elements determined in foodstuffs of plant origin, collected from Swedish market. Indeed the work is really worth and valuable and of course the consumption of these “novel” foods is growing up in the recent years. So these types of studies should be implemented. For this reason I strongly invite the authors to improve this manuscript, to better gather ideas, to develop the sections of the manuscript, and to request the help of a native English speaker for proofreading, because it deserves to be published after several refinements. As it stands, let me say that the manuscript is still in the "draft" stage and needs to be improved.”

Reply:

Thank you for sharing our interest in this study. We have indeed improved the manuscript to better meet the requirements for a full article. We also have had the English ,corrected by using the MDPI English editing service.

“Key issues:”

-        the introduction is really too long and disproportionate to the rest of the work (particularly in comparison of results and discussion)

Reply:

We have shortened the introduction and extended the results and discussion.

-        there are too many errors which make it difficult to read the manuscript smoothly

Reply:

We apologize for not having appropriately corrected these errors in advance. We are sure the manuscript is in better quality as present.

-        the references are indicated in the wrong style and often not indicated at all (I suggest to the authors to re-read with care the journal guidelines)

-        authors should standardize the reference style, check and expand the references used.

Reply:

References are now presented in the correct style as well as thoroughly checked.

-        the style of the conclusion must be changed. It almost feels like a continuation of findings and discussion. You should give first again your objective, then you show your major and most important results and achievements.

Reply:

Conclusions completely rewritten.

-        the presentation of the results and their discussion are insufficient, with such a beautiful dataset you should be able to obtain even more than one article! For example for algae samples the discussion about the content of iAs is only hinted at, while the concentrations of I in these matrices (as well as in the others) it's not really presented. Although I am of the "party" that data often speak for themselves, you cannot think that a dataset alone with an introduction is an article

Reply:

Once again, thank you for sharing our view of the importance of this data. We are well aware of the possibility to evaluate more of the data and present it in more than just one article. However, the beauty is to keep the data in one place and not dived it in fragments, but we do see your point in presenting and discussing more of the results in the text. Three more graphs are added, and the discussion of results extended.

 Some other problems:

-        for all instrumentations reagents standards (Company, City, Country). Standardize and change it in all manuscript.

Reply:

Implemented

- Did you also use “>1 g” of sample for the seaweed? I can't believe it because I know the matrix well and I know that for a complete digestion it is sometimes necessary to add other acids (e.g. hydrofluoric acid...).

Reply:

Yes we did. Though addition of HF may improve recovery for elements associated with refractory phases, those are of little significance in studies focusing on human nutrition. Corresponding explanation is included to the text.

-        the “materials and methods” section is not clear at all.

Reply:

Some modifications have been performed and we hope you will find the present version more clear.

-        check the units of measure of all the text!!! for example: line 332 Are you sure that the limit for cadmium in cereal-based food intended for infants and young children is “0.040 µg/kg”???

Reply:

In present version only mg/kg is used.

-        when using an abbreviation in the text you should introduce it first (e.g. LOD)

Reply:

Implemented

-        it would be nice if you compare your results with other recent studies in the literature.

Reply:

We agree that comparison with previously published data can be beneficial. Such comparison is mentioned for some analytes and matrixes, but detailed comparison will be extremely volumeos and was outside the scope of this study. Moreover, for many many matrixws/elements such results have yet to be published.

-        iodine quantification: Have you had any memory effect problems? same dilution as the other determinations? or even worse recovery problems?

 Reply:

Problems with long was-out times and matrix effects was addressed by special preparation/analysis protocols. Corresponding explanation is included to the text.

Once again, I warmly invite the authors to make efforts to improve the manuscript and then resubmit it.

Reply:

We give you our deepest gratitude for this invitation and hope we fulfil the requirements with this version of the manuscript.

Author Response

Reply to Reviewer 3

We sincerely thank you for the valuable and encouraging comments regarding our manuscript ijerph-2235224 “Cadmium and 73 other elements in foodstuff being major contributors to current dietary exposure of cadmium and in “new food” varieties”. With your help, we have managed to improve it substantially. See our replies  (in the attachment) below after each of your comments/recommendations.

Best regards,

Dr. B Kollander, Dr. Ilia Rodushkin and Ms Birigtta Sundström

Round 2

Reviewer 2 Report

The authors accepted the most part of reviewers' suggestions and modified the manuscript. However there are still some problems

Conclusions

-        -the first sentence of the conclusions is really pretentious .. I want to reassure the authors that there are studies with equally large coverage of matrices and elements (especially because they have not analyzed nor discussed different groups of elements in all matrices). Moreover they also carried out only a basic statistical analysis  and PCA unlike from other studies…. So it should be deleted or changed.

-        -no references should be inserted in this section

In all manuscript: change ruccola in rucola.

To be honest, I still have some doubts about the manuscript .. especially about the discussion..

Also I very much agree with the other reviewer, there are probably too many self-citations, not all of them really appropriate. If you remove these, the remaining references used are rather few…

Even without the tables, I would have done a few more comparisons with the literature.

Author Response

Please, see enclosed file.

Author Response

Please, see enclosed file.
